# Does the openness of the Boundary Shell system influence the sustainable development of the high-tech industry?

**Yiming Shi** [1,2]*, **Qingmei Tan**[1], **Zhi Liu**[3]*, **Ge Yang**[2], **Min Zhang**[2]

**1** School of Economics and Management, Nanjing University of Aeronautics and Astronautics, Nanjing, China, **2** School of Electronic Commerce, Anhui Business College, Wuhu, China, **3** College of Management Engineering, Anhui Polytechnic University, Wuhu, China

* sym727@126.com (YS); liuzhi0551@126.com (ZL)

## Abstract

High-technology industries have gained substantial recognition as pivotal drivers of economic growth and technological advancement in modern society. The imperative of sustainable development in high-tech industries cannot be overemphasized, as it plays a crucial role in enabling long-term growth, fostering innovation, and assuming environmental responsibility. This article presents a study on sustainable development in high-tech industries using Boundary Shell theory. The study investigates the role of the stable and sustainable entropy criterion for the Boundary Shell system of high-tech industries from an entropy balance perspective. It analyzes the upper and lower limits of the Boundary Shell support force. Additionally, it improves the traditional boundary system ratio model to comprehensively and objectively evaluate the sustainable development of high-tech industries. The results illustrate that the Boundary Shell of industrial innovation is stronger than that of external dependency, with a reversed ranking of internal evaluation factor strengths compared to the traditional model. This research integrates reaction-diffusion equations theory with entropy balance equations theory to address sustainability issues in the high-tech industry. We further analyze the sustainable development of the high-tech industry through a Boundary Shell theory perspective to advance sustainability in high-tech industries. Moreover, it provides useful insights into the sustainable development of high-tech industries.

## 1. Introduction

With the rapid development of technology, there has been a huge leap in the scale and quality of manufacturing, and economic development has shifted from high-speed growth to a stage of high-quality development. Along this line, high-tech industries, as engines of economic growth and innovation, have prominent characteristics such as high technical intensity, heavy investment in research and development, wide adoption, and low energy consumption, high-tech industrie play a crucial role in upgrading the national industrial structure and changing the economic growth model. High-tech industries promote economic growth and

**Data Availability Statement:** All relevant data are within the manuscript.

**Funding:** The paper is supported by National Social Science Fund of China (20&ZD127), Anhui

Provincial University Outstanding Young Talents Support Plan (gxyq2021087), Innovation Service Platform of Technical Skills Project in Anhui Business College (2021ZDG07). There was no additional external funding received for this study. The funders had no role in study design, data collection and analysis, decision to publish, or preparation of the manuscript.

**Competing interests:** The authors have declared that no competing interests exist.

development by introducing new technologies, products, and services, and improving production efficiency [1]. In terms of automation and intelligent technology, which are more prominent features of high-tech industries, their advantages in improving production efficiency, reducing resource waste, and environmental pollution are even more obvious. In addition, with the continuous improvement of high-tech industrialization, they provide new types of jobs, create new employment opportunities, and promote social development and stability [2]. However, with the rapid expansion of high-tech industries in recent years, people are paying more and more attention to their sustainability and potential impact on society as a whole. Sustainable development is a global governance issue that involves multiple areas, including economy, society, and environment [3].

In recent years, we are faced with a multitude of complex challenges that require a comprehensive and integrated approach. Sustainable development is a holistic approach to the complex challenges facing our world today, with its core being the pursuit of balance between economic growth, social progress, and environmental protection. As populations grow and economies develop, we are confronted with daunting challenges such as energy scarcity, environmental pollution, and climate change. Only through sustainable development can we achieve the rational use of resources and the protection of our environment. In this digital age, businesses must constantly adapt to changing environmental and societal demands. By adopting sustainable business models and technologies, high-tech industries can reduce resource waste, lower costs, enhance their competitiveness and innovation capabilities [4]. Sustainable development presents new market opportunities for high-tech industries, driving sustained economic growth. By embracing sustainable practices, high-technology industries can reduce their negative effects on the environment and society while promoting sustainable economic growth and technological innovation [5].

The Boundary Shell theory is a theory proposed by Professor Cao Hongxing to study the general laws of system periphery. It emphasizes that the interaction between the system and its environment is carried out through the boundary. In this theory, the boundary is not only the external limit of the system, but also the key area for the interaction between the system and the environment [6]. The theory of Boundary Shell suggests that every system possesses an outer boundary that separates it from the surrounding environment, and the Boundary Shell serves as an intermediary entity that provides protection to the system and facilitates exchange with the environment. Examples of Boundary Shell phenomena can be found in both natural and human systems, such as biological membranes, national borders, atmospheric layers, livestock pens, eggshells, clothing, and internet firewalls, as shown in Fig 1.

In the Boundary Shell system, openness is an important concept. Openness refers to the ability of the system to effectively exchange materials, energy, and information with its environment. This openness enables the system to obtain necessary resources from the environment, as well as discharge waste into the environment, thus achieving the normal operation and development of the system. However, openness does not mean unlimited exchange. Excessive openness may lead to excessive consumption of resources, environmental damage, or even system collapse. Therefore, how to achieve sustainable development of the system while maintaining openness is an important issue facing the Boundary Shell system. According to the Boundary Shell theory, the key to solving this problem lies in finding a balance. This balance should not only ensure that the system can obtain necessary resources from the environment, but also protect the environment from excessive development and destruction [7].

In the perspective of sustainable development, the Boundary Shell theory provides a brand-new way of thinking. By maintaining the security and stability of the system while ensuring interaction and openness with the environment, the Boundary Shell theory contributes to achieving sustainable development of the system. Specifically, by controlling the influx of

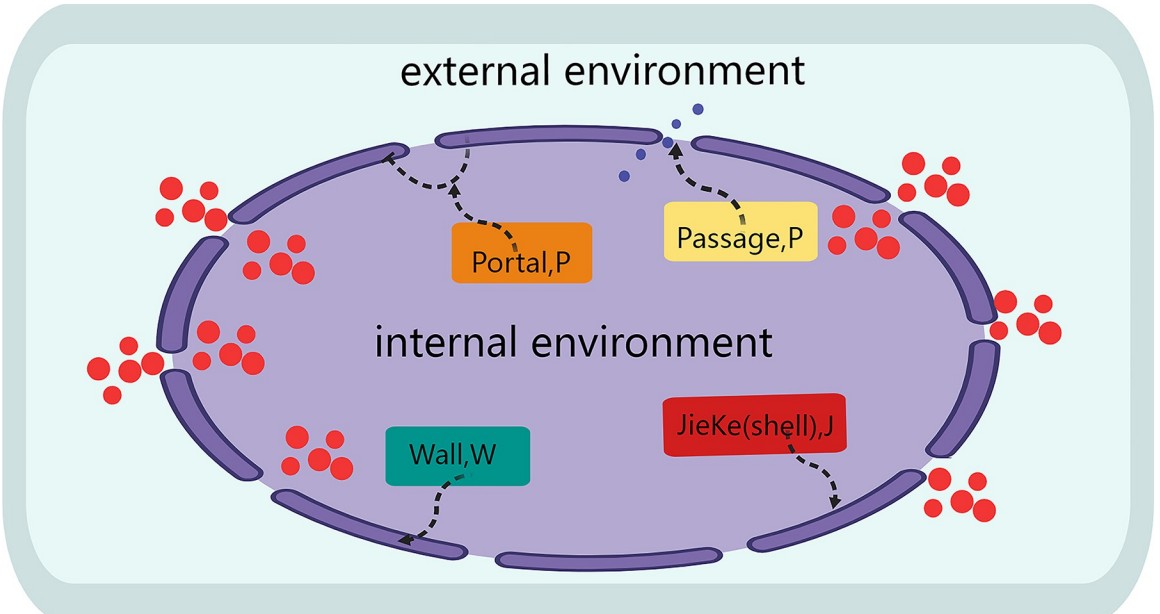

**Fig 1. A schematic diagram of the bound-shell structure.**

substances, energy, and information, the Boundary Shell can prevent the system from being disturbed too much, protecting its core values and resources. At the same time, through open boundary gates, the system can obtain necessary resources and information to support its continuous development. Therefore, we can say that the openness and security of the Boundary Shell system are important conditions for achieving sustainable development.

This article aims to contribute to the field of sustainable development in high-tech industries by employing the Boundary Shell Theory. The study begins with an analysis of the sustainable development of high-tech industries through the lens of Boundary Shell Theory, followed by a discussion on the sustainable stability range of these industries. To further enhance the theoretical framework for understanding the sustainable development of high-tech industries, this paper constructs an improved boundary series ratio model that provides a more precise and comprehensive approach to analyzing the complex interactions between high-tech industries and their environment. This model integrates reaction-diffusion equations theory with entropy balance equations theory to address coordination issues related to green sustainability in high-tech industries. The findings of this study highlight the importance of Boundary Shell Theory in understanding and promoting sustainable development in high-tech industries, and offer new insights for policymakers, practitioners, and researchers seeking to address the complex challenges of sustainable development in the context of high-tech industries.

## 2. Literature review

Our study relates to the literature on the sustainable development of high-tech industry and Boundary Shell theory, the details of which are presented below.

### 2.1. Defining Boundary Shell theory

In the field of intelligent science, the Boundary Shell theory is considered a fundamental theory in China. It provides a comprehensive structure for understanding and managing the

exchange of energy, matter, and information between a system and its environment, with particular emphasis on the role of Boundary Shells. The theory proposes that the Boundary Shell model can represent a system, including "system" (core), "wall" (boundary), "gate" (interface for exchange between the system and the environment), and "environment" (external environment) [8]. We conducted a comprehensive analysis of existing research on Boundary Shell Theory. This theory views the world as an amalgamation of systems that interact with their environment through boundary interactions, which are prevalent in natural and human-made phenomena such as the Earth's atmosphere and biological membranes [9]. Researchers have analyzed the characteristics and spatial reconstruction of Boundary Shells in urban-near-mountain areas, providing guidance for design and planning in these regions from a multidimensional value perspective [10]. Another study investigated the support degree, boundary ratio, and protective capacity of Boundary Shells in high-tech industries, providing a theoretical foundation for design and management in these industries [11]. Additionally, research has highlighted the critical role of Boundary Shells in maintaining a sustainable and healthy ecological environment, emphasizing the need to consider the interconnectedness of ecological, economic, and social factors in sustainable development [12]. Furthermore, researchers have proposed a practical method for evaluating the security of information systems by considering the combined strength of Boundary Shells, providing a useful framework for regulating information exchange between information systems and their environment [13]. Other research focuses on the structure and Boundary Shells of military-civilian industrial alliances in intelligent production and service network systems, providing insights for civil-military integration development [14]. Researchers have also developed a new method for measuring the degree of coupling between different Boundary Shells in intelligent production and service network systems, providing a theoretical basis for optimizing Boundary Shells within the system [15].

Based on the above discussion, the Boundary Shell Theory is proved to be a versatile and multifarious concept with broad applications in diverse fields such as urban planning, industry management, ecological protection, information security, military-civilian integration, transportation safety, complex system networks, and rural tourism safety. The continued development and refinement of Boundary Shell theory have played a crucial role in advancing knowledge and practices towards achieving sustainable development and responsible practices.

## 2.2. Development status of high-tech industry

High-tech industries have a rich history, dating back to the emergence of the electronics industry in Silicon Valley in the 1930 [16]. In recent years, high-tech industries have emerged as key drivers of the global economy, with their core technologies and products playing pivotal roles in fields such as information and communication technology, biotechnology, new materials, and new energy. The US Department of Commerce has been conducting standardized data statistics on high-tech industries since 1965, offering valuable insights into innovation, research and development investment, output, and employment in the industries [17]. These statistics serve as important references for policy-making and industrial development, providing guidance for governments and industry players alike [18]. While the high-tech industry has been a key driver of economic growth and innovation, it also faces challenges in achieving sustainable development.

To address these challenges, it is crucial to gain a comprehensive understanding of the factors that contribute to sustainable competitive advantage and sustainable development in the high-tech industry. Evans [19] investigated the ecological sustainability and the high-tech industry. Izatt [20] examined the challenges to achieving metal sustainability in the high-tech

industry. The study revealed that the increasing demand for metals by the high-tech industry poses a threat to the sustainability of metal resource supply. Rodionov [21] explored the approaches to ensuring the sustainability of industrial enterprises of different technological levels. The study emphasized the importance of a comprehensive sustainability strategy that integrates environmental protection, social responsibility, and economic benefits. Triguero [22] analyzed how financial constraints influence green innovation in high-tech and regulated industries. Pylaeva [23] proposed a new approach to identifying high-tech manufacturing SMEs with sustainable technological development. The study emphasized the importance of sustainable technological development in achieving sustainable competitive advantage. The authors suggested that firms need to adopt sustainable practices in their technological development and collaborate with stakeholders to achieve sustainable development. Zandiatashbar [24] investigated the interplay between the location of high-tech businesses, transportation accessibility, and sustainability. The study revealed that the specialization of the high-tech industry and transportation accessibility are crucial factors in promoting sustainable development. Thus, policymakers must prioritize the provision of appropriate infrastructure and support to foster the growth of the high-tech industry, thereby facilitating sustainable development. Law [25] investigated the motivators and readiness for sustainable development in high-tech manufacturing firms in Hong Kong. The study found that firms need to leverage policy support and technological innovation to promote sustainable development. The policymakers need to provide incentives and support for firms to adopt sustainable practices and promote sustainable development. Rasool [26] explored the policymakers must therefore provide incentives and support to encourage firms to implement sustainable practices, promote the use of renewable energy, and foster innovation in green technology.

The research conducted by the aforementioned scholars highlights the crucial role of sustainable practices in the high-tech industry in achieving sustainable competitive advantage and sustainable development. In contrast, our work uses Boundary Shell theory to study high-tech industry, giving rise to a new research perspective to the sustainable development of high-tech industry.

## 3. Research methods

From the perspective of openness in Boundary Shell systems, this paper proposes a framework for analyzing and determining the upper and lower thresholds of stability in Boundary Shell systems, as shown in Fig 2.

### 3.1. Search strategy and selective database

The rapid advancement of high-tech industries has led to revolutionary changes in various aspects of our lives [27], while Boundary Shell theory provides a framework for understanding the complex relationships between different fields of knowledge. As high-tech industries continue to evolve at an exponential pace, the role of Boundary Shell theory becomes increasingly important in fostering collaboration and cross-pollination between different domains. Together, these two factors form a powerful duo that promises to shape the future of science and technology in significant ways. In this present study, we have employed a bibliometric methodology, drawing upon the comprehensive database of citexs as our foundation. Leveraging the analytical platform of citexs, we embarked on a meticulous exploration of the extensive corpus of literature to uncover valuable insights. By focusing our keyword search on high-tech industries, we identified a total of 3962 articles published between January 2013 and December 2023, with an average annual publication rate of 361 articles. As illustrated in Fig 3, the pinnacle of annual publication volume was reached in 2021 at 467 articles, while the most rapid

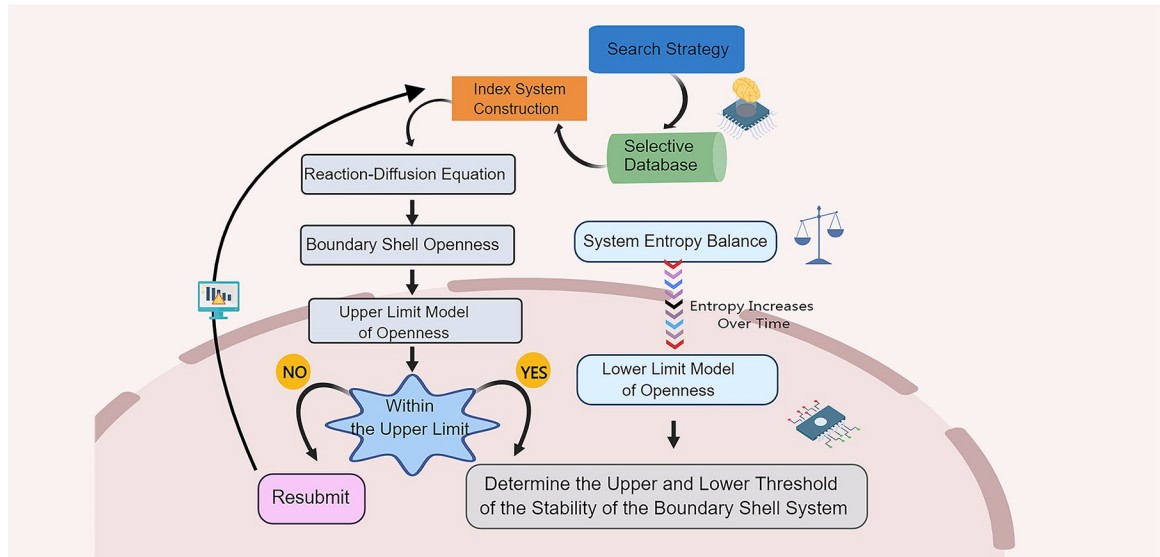

**Fig 2. Analysis flowchart of upper and lower threshold analysis for Boundary Shell system.**

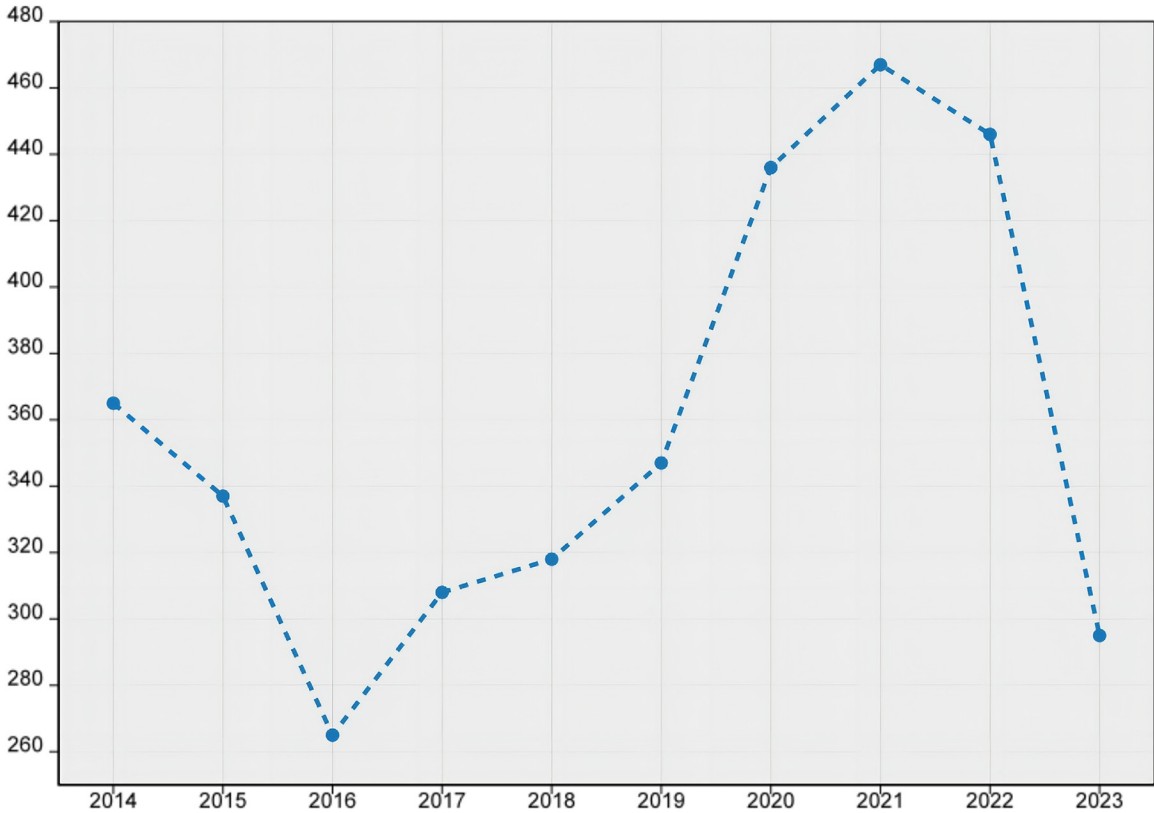

**Fig 3. Annual publication trends of related literature from 2013–01 to 2023–12 high-tech industries.**

**Table 1. 2013–01 to 2023–12 high-tech industry research author analysis.**

| Ranking | Author | Agency & State | Quantity |
|---|---|---|---|
| 1 | Mariacristina Piva | Catholic University of the Sacred Heart, Italy | 8 |
| 2 | Chieh-Peng Lin | National Yang Ming Chiao Tung University, China | 6 |
| 3 | Mark J. Roberts | De Montfort University, United Kingdom | 6 |
| 4 | Peng Huang | University of Mary, United States | 4 |
| 5 | Lara Agostini | Engineering (Italy), Italy | 4 |
| 6 | Anna Nosella | Engineering (Italy), Italy | 4 |
| 7 | M. Muzamil Naqshbandi | Universiti Brunei Darussalam, Brunei | 4 |
| 8 | Daria Honcharenko | Ministry of Economy, North Macedonia | 4 |
| 9 | Shima Hamidi | Johns Hopkins University, United States | 3 |
| 10 | Bo Cowgill | Columbia University, United States | 3 |
| 11 | Matthew J. Higgins | University of Utah, United States | 3 |
| 12 | Piera Centobelli | University of Naples Federico II, Italy | 3 |
| 13 | Benoît Mahy | Université Libre de Bruxelles, Belgium | 3 |
| 14 | Stephen V. Burks | University of Minnesota System, United States | 3 |
| 15 | Qingyuan Zhu | Nanjing University of Aeronautics and Astronautics, China | 3 |
| 16 | Yusheng Xue | Ministry of Education of the People's Republic of China | 3 |
| 17 | Huang Xin-jian | Nanchang University, China | 3 |
| 18 | Svitlana Ishchuk | National Academy of Sciences of Ukraine | 3 |
| 19 | Aistė Miliūtė | Vilnius University, Lithuania | 3 |
| 20 | Jarunee Wonglimpiyarat | Asian Institute of Technology, Thailand | 3 |

growth rate was observed in 2020 at 25.65%. These findings indicate that research within this domain has experienced a remarkable acceleration, currently positioned in a phase of ascending momentum.

Utilizing the analytical platform of Citexs, a bibliometric analysis revealed that between January 2013 and December 2023, the top 20 authors in terms of publication output within the field of high-tech industries research were identified, as shown in Table 1. Notably, Mariacristina Piva emerged as the most prolific author in this domain, having published an impressive total of 8 articles to date. Following closely behind, Chieh-Peng Lin and Mark J. Roberts share second place, each having published 6 articles. Meanwhile, Peng Huang, Lara Agostini, Anna Nosella, M. Muzamil Naqshbandi, and Daria Honcharenko are tied for third place, having each already published 4 articles in this field.

To ensure the reliability of data acquisition, the scope of industry technology is primarily determined by delving into the realms of industry reports, technical documents, relevant literature, and esteemed industry standards pertaining to the target sector. Subsequently, in alignment with the lexicon of high-tech industries, a search expression is formulated, enabling the selection of a database and the acquisition of initial metrics. Finally, through meticulous data cleansing and screening, a cohesive set of effective indicators is obtained. The evaluation index system for innovation capability can be quantitatively screened using R clustering and coefficient of variation methods. This approach involves the application of statistical techniques to identify and prioritize the most relevant and significant indicators within the system. By employing R clustering, similarities or dissimilarities between different indicators can be analyzed and grouped based on their characteristics and relationships [28]. Some scholars have utilized the dynamic capabilities theory to investigate how digital transformation drives the evolution of technological innovation capabilities across three stages [29]. By employing the super-efficient DEA model in conjunction with the Malmquist index, it has been demonstrated that measuring innovation efficiency levels is also an effective approach [30, 31].

In order to ensure the availability and consistency of data, this study has implemented a deliberate sampling approach, focusing on a specific group of 30 provincial-level administrative regions in the People's Republic of China, during the period from 2011 to 2021. This cohort has been selected while excluding Hong Kong, Macao, Taiwan, and Tibet. The data sources for this study primarily consist of official publications, ensuring the reliability and accuracy of the data utilized in the analysis. By adopting this sampling strategy, the study aims to minimize potential errors and inconsistencies, and provide robust and credible findings. including the "China Statistical Yearbook", "China Population and Employment Statistical Yearbook", "China Energy Statistical Yearbook", "China Environmental Statistical Yearbook", "China Science and Technology Statistical Yearbook", as well as provincial statistical yearbooks. Due to the one-year suspension of publication of the "China High-tech Industry Statistical Yearbook" in 2018, Due to the unavailability of statistical data for the year 2017, the collection of statistical data was delayed by a year. The "China High-tech Industry Statistical Yearbook" has already provided specific statistical data on the proportion of high-tech industry export value to manufacturing industry export value. Therefore, no further data processing is required for this data. This will ensure that the analysis is conducted using the most recent and reliable data available, providing accurate and meaningful results.

Before modelling, we employ the symbols and notation shown in Table 2 throughout this paper.

## 3.2. Index system construction

The text details the process of standardizing collected data through the utilization of range normalization methods. This is done to eliminate any potential impact of different unit dimensions that may exist within the data. By using an indicator range of [0,1], the collected data is

**Table 2. Parameters and decision variables.**

| Symbol | Definition | Academic Translation |
|---|---|---|
| $S_V$ | Local Entropy | The entropy within a specific region or area |
| $\vec{J}s$ | Entropy Flow | The entropy change caused by heat, matter or radiation transfer |
| $\delta$ | Local Entropy Generation | The increase in entropy within a specific area |
| V | System | The object of study is an ordered collection of several substances |
| L | System Boundary | The interface between the system and the external environment |
| $dl$ | Boundary Element | A small area used to discretize the system boundary |
| $\vec{n}$ | Unit Vector | A vector perpendicular to the system boundary |
| x | Spatial Coordinates | Mathematical coordinates used to represent different positions |
| $a(x)$ | Entropy Exchange Rate | The rate between the system and the external environment |
| $\vec{J}_e$ | Exchangeable Entropy Flow | The exchanged entropy flow through the system boundary |
| $\rho_u$ | The upper limit of the Open degre | Organizations can achieve this with the external environment. |
| $\rho$ | Open degree | The organization has with the external environment. |
| W | Boundary wall | The wall or interface in a boundary system |
| P | Boundary gate | The gate or interface in a boundary system connects |
| $\vec{J}_{sm}$ | The median on channel P | The median value of various indicators on channel P |
| $\rho_c$ | The lower limit of openness | Minimum openness of the system |
| EJ | Boundary Shell quantity | Amount of matter within the Boundary Shell |
| EI | System quantity | Amount of matter within the system |
| $\eta$ | Original ratio | The original boundary system ratio values |
| $\eta^*$ | Improved Ratio | The improved boundary system ratio values |
| $\omega$ | weights | The weights in a boundary system |

**Table 3. Measurement index system for sustainable development of high-tech industries.**

| First-Level Indicators | Second-Level Indicators | Indicator Description | Weight |
|---|---|---|---|
| Innovation Capability (C) | Sales of new products ($C_1$) | Revenue from sales of new products | 0.1532 |
| | development of new products ($C_2$) | R&D expenditure on new product development | 0.1490 |
| | effective R&D capability ($C_3$) | number of effective patents | 0.2556 |
| Dependence of Industry Innovation Technology (D) | dependence on foreign markets for industry exports ($D_1$) | industry export volume | 0.1956 |
| | dependence on foreign technology for industry innovation ($D_2$) | industry expenditure on technology imports | 0.2466 |

standardized in a manner that ensures comparability and consistency across all units of measurement. The standardization formula is also provided to ensure uniformity and accuracy in the calculation process. By employing this approach, the data is transformed into a format that is more conducive to analysis, allowing for meaningful insights to be drawn from the results.

$$X_{ij}^* = \begin{cases} X_{ij} - \min_j / \max_j - \min_j, \, positive \\ \max_j - X_{ij} / \max_j - \min_j, \, negative \end{cases} \tag{1}$$

The CRITIC method is a powerful multi-criteria decision analysis technique that enables the determination of the relative significance of various criteria. Utilizing the Pearson correlation coefficient, the method computes the intercorrelations among criteria, which serve as a foundation for the calculation of criterion weights. Furthermore, the approach acknowledges the exclusiveness and interdependence between criteria, thereby enhancing its ability to capture the intricacies of the decision-making environment.

$$C_i^* = \sigma_i \sum_{j=1}^{n} (1 - r_{ij}) \quad i = 1, 2, 3, \ldots, m, j = 1, 2, 3, \ldots, n \tag{2}$$

$$W_i = \frac{C_i}{\sum_{i=1}^{m} C_i} i = 1, 2, 3, \ldots, m \tag{3}$$

Based on existing research, the present study has formulated a holistic framework aimed at evaluating the sustainable development of high-tech industries. The framework employs industry innovation capability and external dependence as the primary indicators, which have been selected after careful consideration of their significance. By utilizing these indicators, the framework provides a comprehensive and multifaceted approach to measuring the sustainable development of high-tech industries. To obtain the weight values of each indicator, the original data underwent standardization before being input into the CRITIC calculation formula. The specific index system utilized is presented in Table 3.

## 3.3. Upper limit model of openness of Boundary Shell system

The exchange of substances within and outside the boundary membrane system is influenced by the degree of openness of the membrane wall. To maintain the integrity of the system, the degree of openness of the boundary membrane must remain below 1. This suggests the presence of an upper threshold that should not be surpassed. If the degree of openness exceeds this threshold, it may result in an excessive flow of information or energy within the system, leading to confusion and denaturation of the system's properties. To investigate this threshold, this paper aims to build on previous research by discussing the determination of an upper limit for the boundary membrane system. In doing so, the paper will explore the reaction-diffusion equation as a

mathematical model for describing the movement of substances, accounting for both diffusion and chemical reactions (Cao, 1995) [6]. The equation can be expressed as follows:

$$\frac{\partial \rho_j}{\partial t} = D_j \nabla^2 \rho_j + f_j(\{\rho_j\}) \tag{4}$$

Based on the reaction-diffusion equation, Cao derived the upper bound of the Boundary Shell openness, with the specific model as follows:

$$\int_0^p \frac{\partial \rho_j}{\partial_t} dl = \int_0^p D_j \nabla_p^2 \rho_j dl + \int_0^p f_j\{\rho_j\} dl \tag{5}$$

The external components of the Boundary Shell system are composed of the boundary wall and the boundary gate. The boundary wall is specifically engineered to impede the flow of mass, resulting in a zero-diffusion term for the Boundary Shell system. Let $R_j = \int_0^p \rho_j dl$, obtain the openness of the Boundary Shell:

$$\rho = \frac{p}{l} = \frac{(\frac{\partial R_j}{\partial_t} - D_j \nabla_p^2 R_j)}{l f_j(\{\rho_j\})} \tag{6}$$

As shown in the above equation, the necessary condition for maintaining the properties of the Boundary Shell system is $\rho < \rho_u$, otherwise, if the openness of the boundary shell system is too high, it may cause system degeneration problems.

## 3.4. Lower limit model of openness of Boundary Shell system

The principle of system entropy balance postulates that in a closed system, the overall entropy remains stable or increases over time unless there is an external influx of energy or matter. Essentially, a system's entropy will tend towards a state of maximum disorder or randomness unless counteracted by some sort of energy input or organization. This concept is fundamental in thermodynamics and holds significant implications for understanding physical, chemical, and biological systems. Cao employed the entropy balance equation to develop a model for the openness of boundary-shell systems. In particular, the model assumes that the boundary-shell wall is insulating and impedes any transmission of matter, energy, or information, making the boundary gates the sole interface for interaction between the system's interior and exterior. Assuming an entropy inflow of $\overrightarrow{J}_{in}$ and an entropy outflow of $\overrightarrow{J}_{ex}$ for the boundary-shell system, The equation is as follows:

$$\frac{dS}{dt} = -\int_P \overrightarrow{n} \cdot \overrightarrow{J}_{ex} dl + \int_P \overrightarrow{n} \cdot \overrightarrow{J}_{in} dl + \tilde{P} \tag{7}$$

Clearly, based on the second law of thermodynamics, for a boundary-shell system to maintain stable development. Herein, due to this, it is necessary to satisfy the condition of $\tilde{P} \geq 0$, resulting in equation as:

$$\rho > \frac{\tilde{P}}{(\overrightarrow{n} \cdot \overrightarrow{J}_{sm})l}(p = \rho l) \tag{8}$$

## 3.5. Boundary system ratio model of Boundary Shell system

The Boundary Shell theory posits a definition of the boundary system ratio as the ratio of the Boundary Shell quantity to the system quantity, which is characterized by a value of $\gamma = Q_J/Q_I$.

The conventional boundary system ratio model is a rudimentary representation of a ratio that disregards the full scope of indicator data. This insufficiency is remedied in this study by introducing the concept of mass weight to the initial boundary system ratio model. The argument is made that for determining the boundary system ratio for the protective Boundary Shell of the high-tech industry, it is necessary to consider the weight of each indicator.

By utilizing the boundary system ratio criterion, the protective power of the Boundary Shell can be quantified. To this end, the total Boundary Shell quantity of the high-tech industry is represented by $E_J$, the total system quantity is represented by $E_I$. The weight of each mass element is represented by $\omega_n$, and these parameters can be used to derive an improved boundary system ratio model.

$$\eta = \frac{E_J}{E_I}\omega = \frac{1}{n}\sum_{i=1}^{n}\frac{E_{Ji}}{E_{Ii}}\omega_i \tag{9}$$

By doing so, we provide a comprehensive understanding of the complexities inherent in the study of Boundary Shell systems. The Upper limit model serves as a cornerstone for exploring the upper bounds of openness within these systems, while the Lower limit model sheds light on the lower limits that govern their behavior. Furthermore, the refined Boundary system ratio model offers a nuanced perspective on the interplay between various factors that influence the dynamics of Boundary Shell systems.

## 4. Empirical analysis

### 4.1. Data source and samples

After conducting an exhaustive analysis of the research findings in the world, our attention now turns to the global landscape of high-tech research. The top 99 countries/regions in terms of publication output in the field of high-tech industries research are depicted in Fig 3. The country/region with the highest publication output in this field is China, with a total of 950 articles (accounting for 23.98%), followed by Russia (575 articles, or 14.51%) and the United States of America (486 articles, or 12.27%), which occupy second and third place respectively. As depicted in Fig 4, the data analysis results obtained from the CiteXs platform reveal significant disparities in the research output of various countries/regions within the realm of high technology. This disparity in research productivity underscores the diverse approaches and strategies adopted by different nations in their pursuit of technological advancements. The analysis provides invaluable insights into these regional variations, enabling us to gain a deeper understanding of the global dynamics shaping the field of high technology.

In this section, our research spotlight centers on theIn this section, our research spotlight centers on the high-tech industry in China, a domain that has garnered widespread attention on a global scale. Driven by the rapid development of the Chinese economy and propelled by technological innovations, the country's high-tech industry has emerged as a significant player in the global arena.

### 4.2. Descriptive statistical analysis

To delve deeper into the trends and challenges shaping the sustainable growth of China's high-tech sector, we employ the upper and lower bound models constructed in the preceding text. we conduct numerical studies on the performance of the revised value proposed in this paper. In the parameter designs, we set the innovation capability and external dependency of China's high-tech industries between 2011 and 2021 are computed using the boundary system ratio model. This arrangement ensures that we operate within the region of feasibility. The

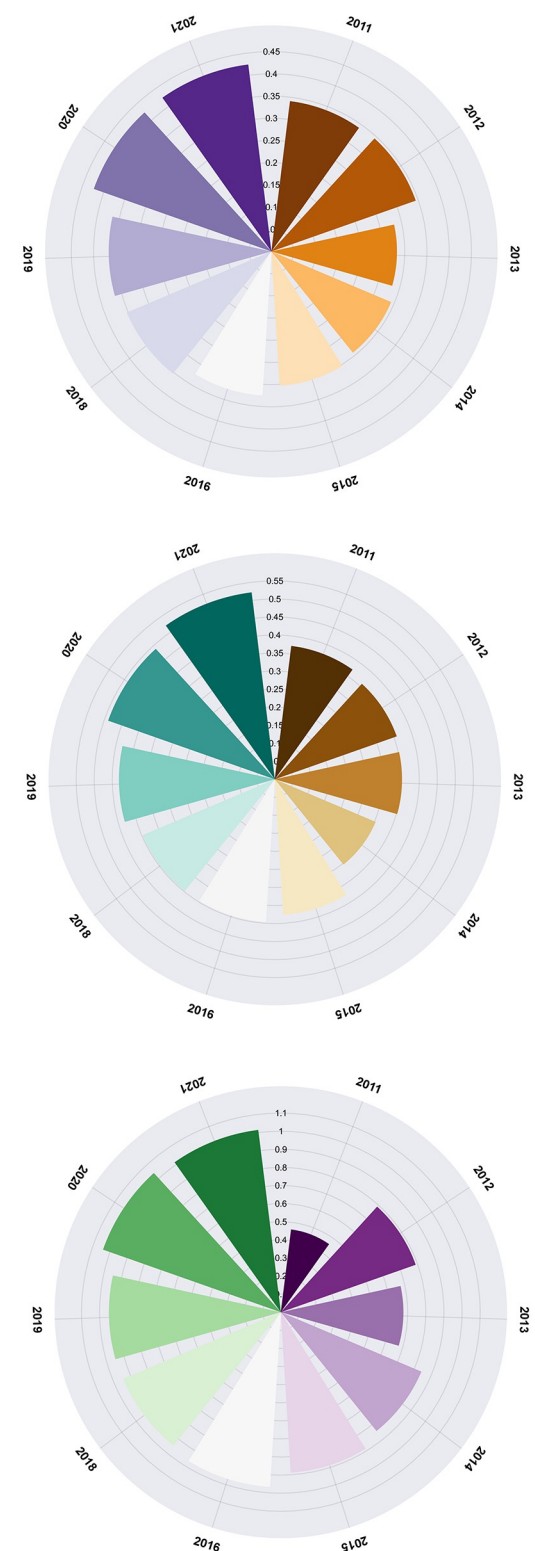

**Fig 4. The ratio of C1- C3 in 2011–2022.**

**Table 4. The boundary system ratio values of high-tech industries.**

| | $\gamma_{C_1}$ | $C_{1\eta}$ | $\gamma_{C2}$ | $C_{2\eta}$ | $\gamma_{C3}$ | $C_{3\eta}$ | $\gamma_{D_1}$ | $D_{1\eta}$ | $\gamma_{D2}$ | $D_{2\eta}$ |
|---|---|---|---|---|---|---|---|---|---|---|
| 2011 | 0.2543 | 0.3410 | 0.2714 | 0.3725 | 0.3153 | 0.4605 | 0.2672 | 0.3646 | 0.1989 | 0.2483 |
| 2012 | 0.2561 | 0.3443 | 0.2636 | 0.3580 | 0.4410 | 0.7889 | 0.2561 | 0.3443 | 0.2047 | 0.2574 |
| 2013 | 0.2203 | 0.2825 | 0.2606 | 0.3524 | 0.4035 | 0.6764 | 0.2752 | 0.3797 | 0.1679 | 0.2018 |
| 2014 | 0.2263 | 0.2925 | 0.2328 | 0.3034 | 0.4572 | 0.8423 | 0.2751 | 0.3795 | 0.1823 | 0.2229 |
| 2015 | 0.2325 | 0.3029 | 0.2734 | 0.3763 | 0.4702 | 0.8875 | 0.2581 | 0.3479 | 0.1566 | 0.1857 |
| 2016 | 0.2452 | 0.3249 | 0.2838 | 0.3963 | 0.4915 | 0.9666 | 0.2634 | 0.3576 | 0.1981 | 0.2470 |
| 2018 | 0.2610 | 0.3532 | 0.2846 | 0.3978 | 0.4859 | 0.9451 | 0.2713 | 0.3723 | 0.1454 | 0.1701 |
| 2019 | 0.2679 | 0.3659 | 0.3015 | 0.4316 | 0.4868 | 0.9486 | 0.2542 | 0.3408 | 0.1590 | 0.1891 |
| 2020 | 0.2975 | 0.4235 | 0.3285 | 0.4892 | 0.5099 | 1.0404 | 0.2582 | 0.3481 | 0.1968 | 0.2450 |
| 2021 | 0.2978 | 0.4241 | 0.3430 | 0.5221 | 0.5036 | 1.0145 | 0.4460 | 0.8051 | 0.2198 | 0.2817 |
| $\overline{X}$ | 0.2559 | 0.3455 | 0.2843 | 0.4000 | 0.4565 | 0.8571 | 0.2825 | 0.4040 | 0.1830 | 0.2249 |

protective boundary layer and enclosure boundary layer boundary system ratio values for China's high-tech industries are shown in Table 4. Using $\gamma$ to represent Ratio. In this study, we utilized the boundary system ratio model, along with MATLAB software, to precisely assess the innovation capability and external dependency of high-tech industries. We further enhanced the original boundary system ratio values by incorporating weights.

The performance indicators of the innovation capability boundary system are visually represented in chart, as depicted in Fig 4.

As depicted in Fig 4, the three metrics under consideration are the new product sales capability boundary ratio (C1), new product development capability boundary ratio (C2), and effective R&D capability boundary ratio(C3), both before and after enhancement. The graph spans a period from 2011 to 2022, during which the new product sales capability boundary ratio and new product development capability boundary ratio fluctuated between 0.28 to 0.43, and 0.30 to 0.53, respectively. Despite the aforementioned values remaining securely within the acceptable boundaries of the established system, it is important to note that the fluctuating effective R&D capability boundary ratio exhibited significant deviations between 0.46 and 1.02, exceeding the upper limit of the boundary system in both 2020 and 2021. This highlights the need for further investigation into the factors contributing to these fluctuations and potential measures to mitigate their impact on the overall performance of the system. It is crucial to analyze the external factors that may have influenced the fluctuations in the R&D capability boundary ratio. These could include changes in market conditions, advancements in technology, or shifts in consumer preferences. By identifying and understanding these external influences, organizations can better anticipate and adapt to the evolving landscape, ensuring that their R&D efforts remain aligned with market demands.

This revelation indicates that the Fig 4 reveals that the ranges of fluctuation for the new product sales capability boundary ratio and new product development capability boundary ratio are quite similar, implying that the high-tech industries' capabilities in both areas are relatively evenly matched. However, the effective R&D capability boundary ratio exhibits a wider range of fluctuations, indicating a certain degree of imbalance between R&D investment and effectiveness. Therefore, there is an imperative need to strengthen R&D management and optimize the investment structure in this area, to enhance the efficiency and effectiveness of R&D endeavors. To achieve this, it is crucial to adopt a comprehensive approach that encomTo achieve this, it is crucial to adopt a comprehensive approach that encompasses various aspects of R&D management. Firstly, organizations must establish clear goals and objectives for their

R&D activities, aligning them with the overall business strategy. This will help ensure that resources are allocated efficiently and effectively towards projects that have the highest potential for success. In addition to these measures, strengthening R&D management and optimizing investment structures are imperative for enhancing the efficiency and effectiveness of R&D endeavors. By establishing clear goals, fostering collaboration, investing in talent, engaging with external partners, and continuously monitoring performance, organizations can position themselves for sustained success in the rapidly evolving landscape of research and development.

In line with this, our investigation takes it a step further by incorporating two key indicators of the external dependence boundary system: the ratio of industrial exports' external dependence and the ratio of industrial innovation technology's external dependence, as depicted in Fig 5.

As shown in Fig 5, the trends of these indicators are observed over a time frame spanning from 2011 to 2021. During this period, the ratio of industrial exports' external dependence fluctuated between 0.34 and 0.81, while the ratio of industrial innovation technology's external dependence ranged from 0.17 to 0.29. It is noteworthy that both of these indicators remained within the upper and lower limits of the boundary system's supportability. In calculating the original boundary system ratio model, the authors separately calculated the innovation capability boundary system ratio $\overline{C}_\eta = 0.5342$ and external dependency boundary system ratio $\overline{D}_\eta = 0.3145$ of high-tech industries. The study reveals that the protective capabilities of the two boundary layers are not equal, and the innovation capability boundary layer exhibits a stronger ability to safeguard against potential threats or risks. The study findings highlight a significant disparity in the protective capabilities of the two boundary layers, with the innovation capability boundary layer demonstrating a markedly stronger ability to safeguard against potential threats or risks. This suggests that organizations must prioritize and invest in cultivating a culture of innovation if they aim to effectively navigate the complex and dynamic business landscape of today. Based on Fig 5, it can be observed that innovation plays a critical role in safeguarding organizations against potential threats and risks. By prioritizing innovation, fostering a culture of openness and collaboration, adopting a flexible approach, and investing in R&D efforts, organizations can strengthen their protective capabilities and position themselves for long-term success in an increasingly competitive business landscape.

## 4.3. Comparison of the model before and after refinement

With the aim of enhancing the precision of assessing the innovation capability and external dependency of high-tech industries, the authors employed an advanced boundary system ratio model to evaluate the protective capacities of two distinct boundary layers. Table 5 presents a comparative analysis of the boundary system ratio model before and after improvements.

As depicted in Fig 6, the boundary system ratio values for depicted in Fig 7, the boundary system ratio values for each indicator were calculated using an pre-boundary system model. The original model's indicators were ranked in descending order of potency as follows: effective research and development patent rate (C3 = 0.8574), industrial export external dependency ratio (D1 = 0.4040), new product development ratio (C2 = 0.4000), new product sales ratio (C1 = 0.3455), and industrial technology external dependency ratio (D2 = 0.2249). Upon further analysis of the computed boundary system ratio values, it becomes apparent that the per capita effective R&D patent ratio holds the highest position among the indicators. This suggests a strong emphasis on innovation and technological advancement within the industry, which is crucial for maintaining a competitive edge in today's rapidly evolving market landscape.

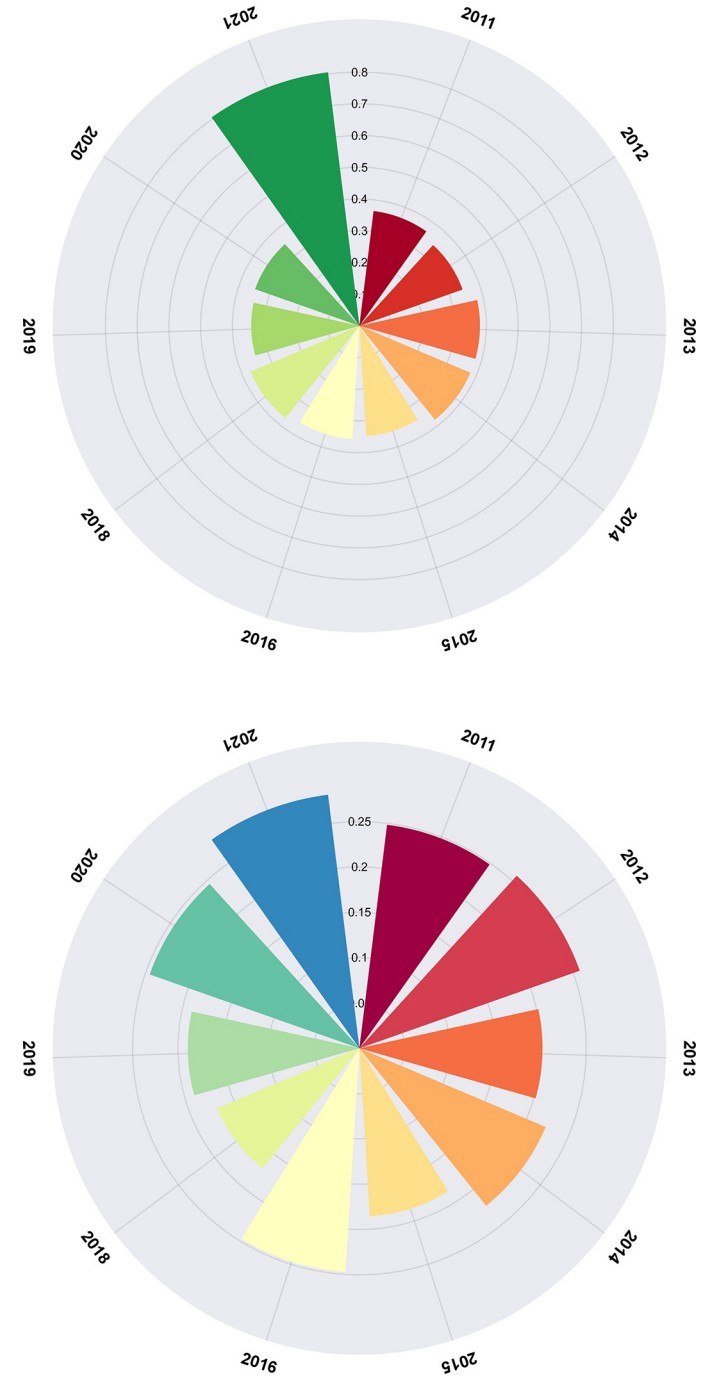

**Fig 5. Ratio of bounds between D1 and D2 from 2011 to 2022.**

The implementation of the refined boundary system ratio model has resulted in a significant shift in the ratios, as illustrated in Fig 7. The indicator values have been arranged in descending order of strength, beginning with per capita effective R&D patent rate (C3), followed by industrial export external dependency ratio (D1), new product development ratio (C2), industrial technology external dependency ratio (D2), and culminating with new product sales ratio (C1). This analysis underscores the stark disparities in evaluation outcomes before

**Table 5. A comparative analysis of the boundary system ratio model before and after modification.**

|  | $C_1$ | $C_2$ | $C_3$ | $D_1$ | $D_2$ |
|---|---|---|---|---|---|
| $\eta$ | 0.3455 | 0.4000 | 0.8571 | 0.4040 | 0.2249 |
| $\omega$ | 0.1532 | 0.1490 | 0.2556 | 0.1956 | 0.2466 |
| $\eta^*$ | 0.0529 | 0.0596 | 0.2191 | 0.0790 | 0.0555 |

In the table, η represents the original boundary system ratio values, ω denotes the weights, and η* indicates the improved boundary system ratio values. This enhanced model enabled more precise calculation of the protective capabilities, thus providing greater insight into the strengths and weaknesses of each boundary layer. The results showed that the innovation capability boundary system ratio was $\overline{C}_{\eta^*} = 0.1105$ and the external dependency boundary system ratio was $\overline{D}_{\eta^*} = 0.0673$.

and after the enhancement of the model. Notably, there has been a marked alteration in the ranking of advantages, with the new product sales rate now surpassing the industrial technology external dependency ratio. This may indicate a shift in industry priorities, as organizations strive to maintain their leading position and competitive edge, they may need to place greater emphasis on developing innovative products that can drive growth and capture market share. The analysis reveals the profound impact of the refined boundary system ratio model on the overall shell structure of the high-tech industry.

As illustrated in Fig 8, a comparative analysis of the two sets of indicator values before and after modification is presented within the same chart. It can be observed that the innovation

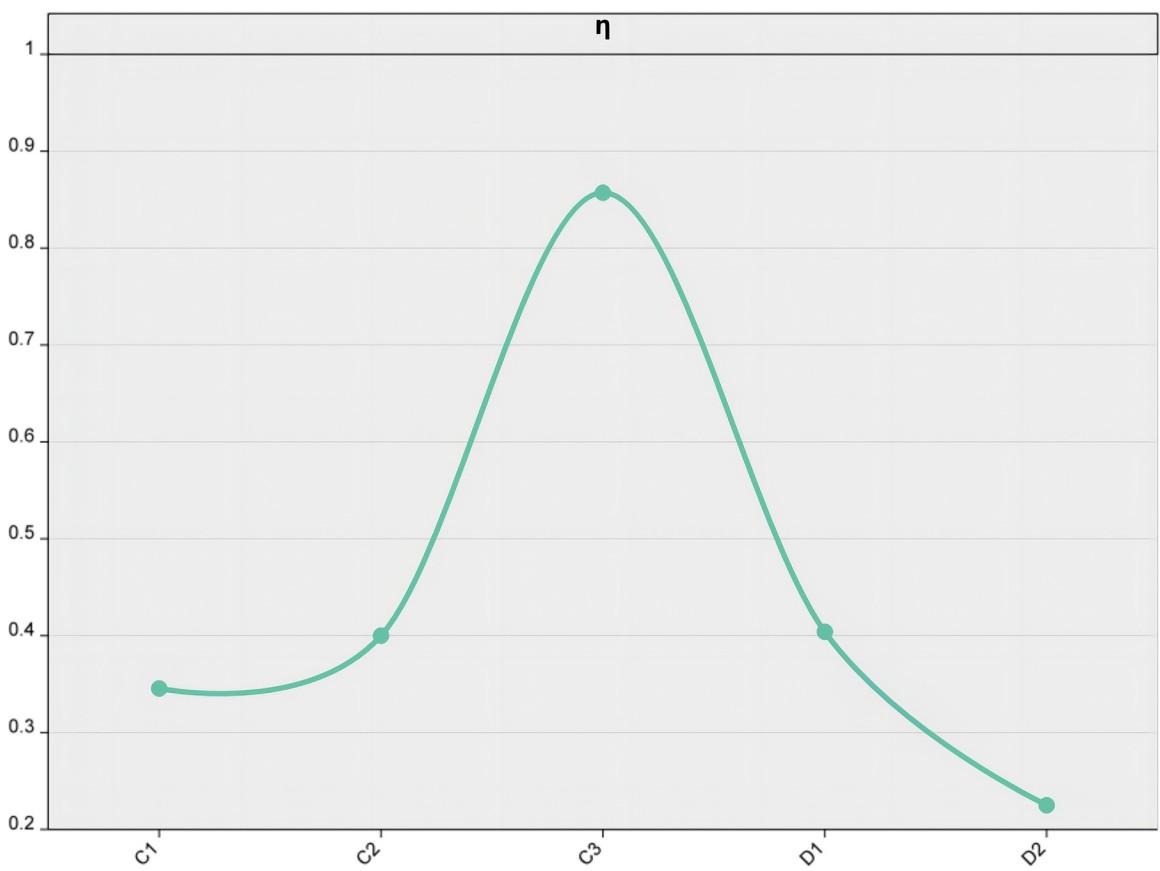

**Fig 6. Trend change before improvement in Boundary System.**

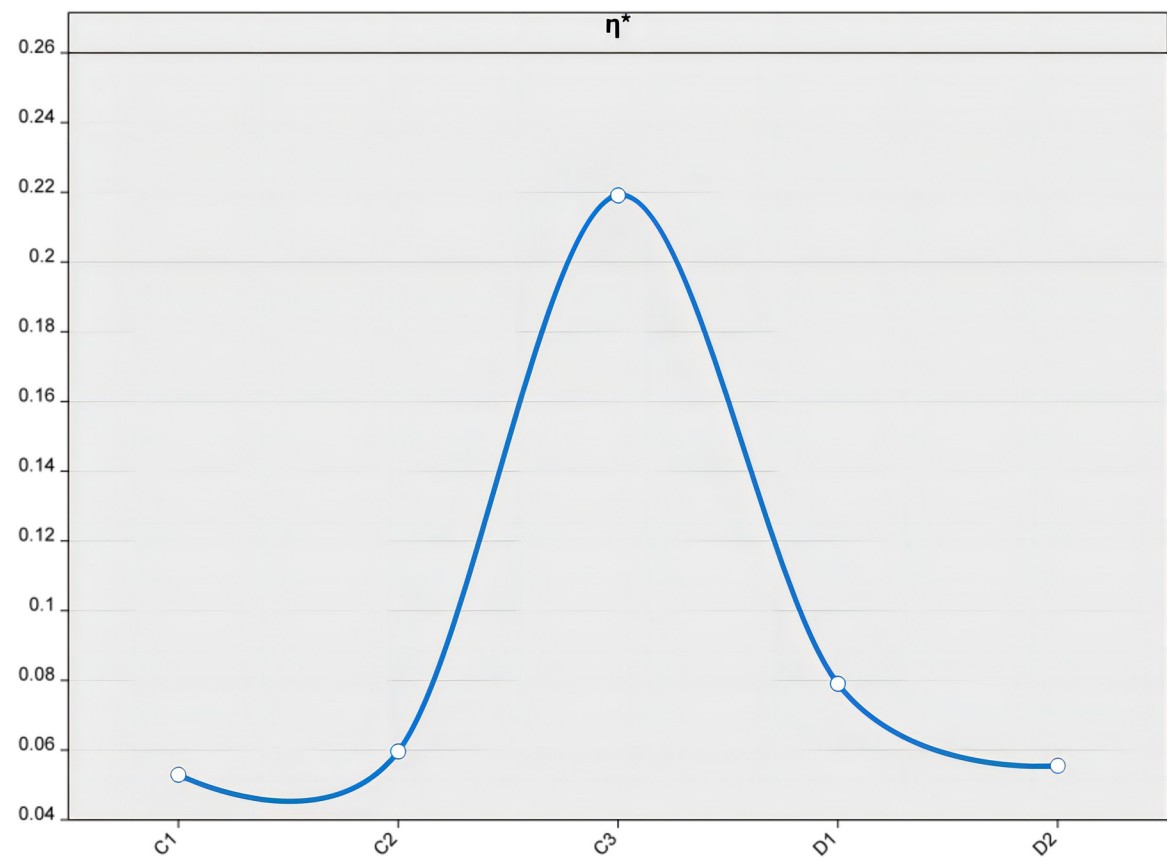

**Fig 7. Trend change after improvement in Boundary System.**

capability boundary system prior to modification was relatively unstable during this period, resulting in uncertainty and risk in enterprise innovation activities. However, following model modification, the fluctuation amplitude of all indicators has significantly decreased, thereby effectively controlling their volatility. Furthermore, the stability and reliability of the innovation capability boundary system have been notably enhanced, reducing the uncertainty and risk associated with enterprise innovation activities. This out come substantiates the effectiveness of the proposed methodology.

The integration of indicator weights into the refined boundary system ratio model affords a more exhaustive and unprejudiced appraisal of the sustainable development capabilities of high-tech industries, relative to the original model. The study reveals that the employment of boundary system ratios constitutes a more accurate and dependable methodology for evaluating the innovation capacity and external dependency of high-tech industries. The findings of this investigation can furnish invaluable insights to policymakers and stakeholders, thereby empowering them to formulate well-informed decisions and devise effective policies.

## 5. Conclusions remarks and implication

### 5.1. Conclusions

Through a series of investigations conducted in this paper, we aim to explore whether the openness of Boundary Shell systems impacts the sustainable development of high-tech industries. Throughout this process, relevant data has been collected and subjected to meticulous

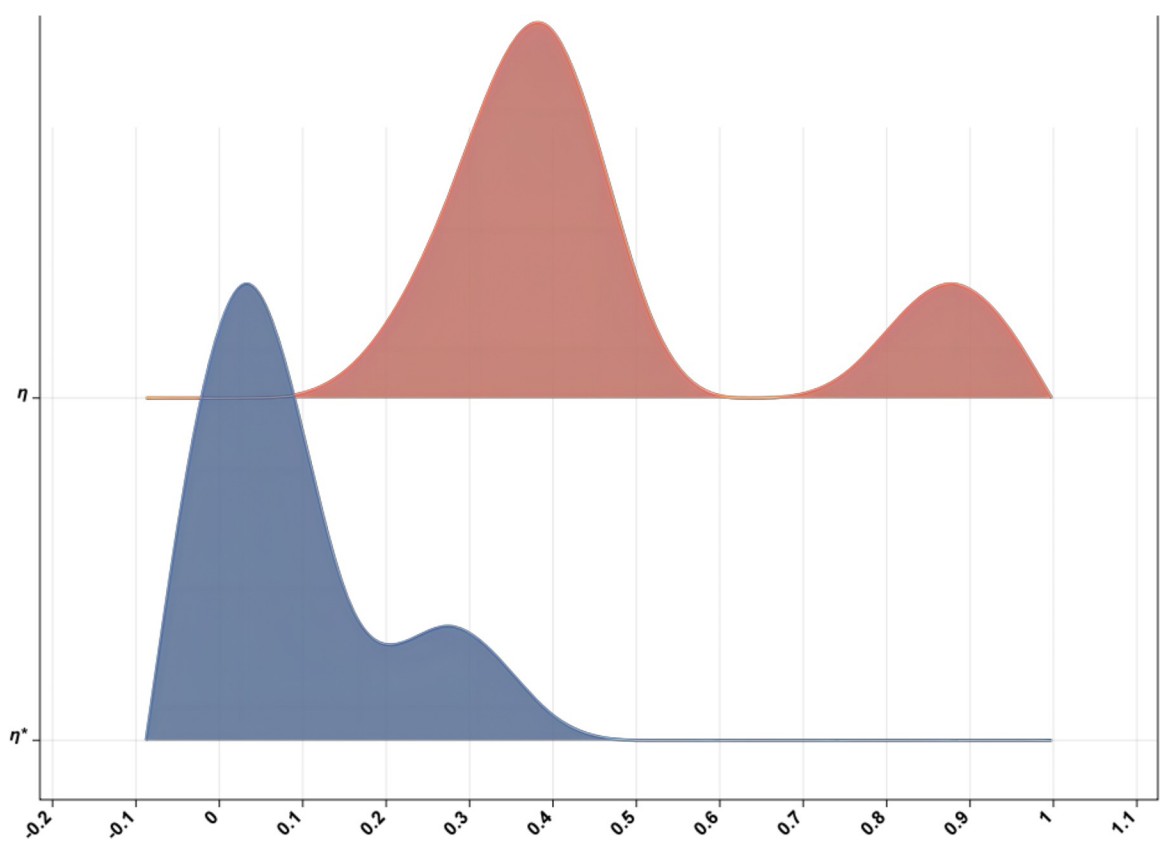

**Fig 8. Comparison diagram before and after the improvement.**

analysis. In this section, we will provide a comprehensive summary and discussion of our research findings with the purpose of gaining a deeper understanding of the factors and underlying mechanisms that drive the sustainable development of high-tech industries through the lens of Boundary Shell theory. The main results are summarized as follows:

1. This study delves into the complex interplay between the quantity of Boundary Shell and internal system. By drawing upon entropy equilibrium principles, a comprehensive examination is conducted to establish a rigorous benchmark for the stable and enduring entropy of the Boundary Shell system within cutting-edge industries. This investigation offers invaluable insights into the upper and lower thresholds that define the supportive force necessary to maintain the integrity of the Boundary Shell. The analysis presented herein highlights the critical role played by entropy equilibrium principles in understanding the behavior of the Boundary Shell system. Through a meticulous examination of the relationship between the quantity of Boundary Shell and internal system, this study provides a robust framework for assessing the stability and longevity of entropy within such systems.

2. After the refinement of the boundary system ratio model, a significant shift in the prioritization of indicator industries has occurred. The rate of new product sales now surpasses the external dependency ratio of industrial technology, reflecting heightened attention towards the development of innovative products that can drive growth and capture market share. This transformation is underscored by the refined boundary system ratio model's ability to provide a more nuanced assessment of high-tech industries' sustainable development capabilities. By integrating indicator weights and considering the varying significance of

different indicators, this revised model offers a more comprehensive and unbiased evaluation. This shift in priorities signifies an industry-wide recognition of the critical role that innovation plays in achieving sustainable development in high-tech industries. The refined boundary system ratio model serves as an invaluable tool for decision-makers, stakeholders, and researchers in evaluating and guiding the sustainable development of high-tech industries. Through the introduction of the refined shell model, which accounts for the varying importance of different indicators, a more detailed assessment can be achieved, providing a more accurate representation of the overall sustainability landscape.

3. The boundary system ratio model has undergone a continuous refinement process to enhance its capacity in evaluating the sustainable development capabilities of high-tech industries. This refinement involves incorporating indicator weights into the model, which plays a critical role in this enhancement. By assigning appropriate weights to each indicator, a more comprehensive and unbiased assessment can be achieved compared to the original model. The integration of indicator weights into the boundary system ratio model enables it to capture the relative importance of different factors that promote sustainable development in the high-tech industry. By introducing the modified boundary system ratio model, which considers the varying significance of different indicators, a more nuanced evaluation can be achieved, providing a more accurate representation of the overall sustainability landscape. The modified model serves as a valuable tool for decision-makers, stakeholders, and researchers in assessing and guiding the sustainable development of high-tech industries. It provides a comprehensive framework for understanding and optimizing the long-term prospects of these sectors.

4. Furthermore, our findings have been juxtaposed with alternative theoretical constructs, revealing that notwithstanding certain methodological divergences [32], our model proffers enhanced precision and discernment in the appraisal of sustainable development within high-tech industries [33]. In contradistinction to theoretical forecasts [34], our paradigm, by initiating from the perspective of Boundary Shell Theory and employing the auxiliary and primary support levels of the boundary system as analytical vehicles, achieves a more veracious representation of the actual market dynamics and technological progression trends. This comparative analysis not only corroborates the efficacy of our model but also delineates novel trajectories for future investigative endeavors.

## 5.2. Policy implication

In light of the aforementioned discoveries, the following research implications are suggested.

1. Collaboration is the key to unlocking the full potential of high-tech industries and ensuring their long-term success. Establishing a robust network of collaboration is crucial for promoting the sustainable development of these industries. By fostering close collaboration among policymakers, industry stakeholders, and researchers, a comprehensive understanding of the complex dynamics governing the relationship between Boundary Shells and internal systems can be ensured. Such collaboration can drive interdisciplinary research and knowledge sharing, thereby accelerating innovation and development. Additionally, collaboration can facilitate the rational allocation and effective utilization of resources, avoiding duplication of efforts and waste. In high-tech industries, each stakeholder brings unique expertise and resources, and through collaboration, they can achieve complementary advantages, jointly address challenges, and achieve common goals.

 

2. The principle of entropy equilibrium provides a valuable framework for evaluating the stability and lifespan of high-tech industry Boundary Shell systems. Prioritizing the use of entropy equilibrium principles in evaluating the stability and lifespan of these systems and incorporating these principles into policy formulation and decision-making processes allows policymakers to gain valuable insights into the behavior of these systems and make more informed choices. By adopting proactive approaches to managing entropy levels, policymakers can help ensure that high-tech industries remain resilient and able to adapt to an ever-changing environment.

3. The openness of Boundary Shell systems plays a critical role in promoting the sustainable development of high-tech industries. Policymakers should prioritize initiatives that foster innovation and the adoption of cutting-edge technologies within these domains. This can be achieved through targeted investments in research and development and by providing incentives for companies to explore new ideas and approaches. Additionally, policies should be developed to encourage collaboration among industry participants, academia, and research institutions to facilitate the exchange of knowledge and expertise. By promoting a culture of innovation and continuous improvement, high-tech industries can contribute to sustainable development by developing more efficient, environmentally friendly, and socially responsible products and services.

## 5.3. Limitations

The limitations and future prospects of this study are as follows: (1) This study employs a quantitative approach to investigate the impact of openness in Boundary Shell systems on the sustainable development of high-tech industries, which may not fully capture the complex and multifaceted policy implications present in today's intricate realities. While this method has its merits, it is limited in providing a comprehensive understanding of the nuanced dynamics at play. To address this limitation, future research could incorporate qualitative methods to gain deeper insights into the complexity of policy impacts. (2)The concept of Boundary Shell systems in this study refers to the interconnected networks of actors, institutions, and policies that shape the development of high-tech industries. This theoretical framework provides a useful lens for examining the impact of openness on sustainable development in this context. However, it is important to recognize that there are alternative theories and perspectives that could offer additional insights into this complex issue. Future research could benefit from incorporating diverse theoretical perspectives to provide a more comprehensive understanding of the factors that influence sustainable development in high-tech industries.(3)The experimental data sample in this paper focuses solely on policy impacts within specific countries and regions, thereby limiting its applicability to other geographical contexts. In order to enhance the generalizability of the findings, future research should strive to expand its scope and include a broader range of countries and regions. By doing so, researchers can develop a more holistic understanding of policy impacts that transcend national boundaries and have greater significance for global decision-making.

## Author Contributions

**Conceptualization:** Yiming Shi, Qingmei Tan.

**Data curation:** Zhi Liu.

**Formal analysis:** Ge Yang.

**Investigation:** Min Zhang.

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
