## [Decision Letter · Decision Letter 0]

2 Nov 2023

PONE-D-23-29413Does the openness of the Boundary Shell system influence the sustainable development of the high-tech industry?PLOS ONE

Dear Dr. Shi,

Thank you for submitting your manuscript to PLOS ONE. After careful consideration, we feel that it has merit but does not fully meet PLOS ONE’s publication criteria as it currently stands. Therefore, we invite you to submit a revised version of the manuscript that addresses the points raised during the review process.

PLOS ONE requires two substantial reviews and both reviewers recommended a "Major Revision." Please respond to the reviewers as my final decision will be a function of such responses and PLOS ONE’s publication criteria. Please submit your revised manuscript by Dec 17 2023 11:59PM. If you will need more time than this to complete your revisions, please reply to this message or contact the journal office at plosone@plos.org. Please include the following items when submitting your revised manuscript:A rebuttal letter that responds to each point raised by the academic editor and reviewer(s). You should upload this letter as a separate file labeled 'Response to Reviewers'.A marked-up copy of your manuscript that highlights changes made to the original version. You should upload this as a separate file labeled 'Revised Manuscript with Track Changes'.An unmarked version of your revised paper without tracked changes. You should upload this as a separate file labeled 'Manuscript'.

We look forward to receiving your revised manuscript.

Kind regards,

Voxi Heinrich Amavilah, Ph.D.

Academic Editor

PLOS ONE

Journal Requirements:

Yiming Shi conceptualized the study. Qingmei Tan and Yiming Shi were responsible for the study methodology. Data curation and analysis were performed by Qingmei Tan. Zhi Liu performed the investigations, and the resources were provided by Ge Yang. The first draft of the manuscript was written, reviewed, and edited by Ge Yang and Min Zhang. All authors commented on previous versions of the manuscript. All authors read and approved the final manuscript.

I have read the journal's policy and the authors of this manuscript have the following competing interests

Additional Editor Comments:

Reviewer # 1

There is an incorrect citation in line 120, and the font inconsistency in line 163.

The introduction of the paper does not effectively introduce the topic and feels abrupt. It is recommended to include some background information in the introduction, such as the importance of the high-tech industry and the significance of sustainable development, to provide a better foundation for the subsequent discussion.

The theoretical framework section lacks detailed exposition and explanation. It is suggested to delve deeper into the relationship between the Boundary Shell system's openness and sustainable development, and provide a clearer theoretical framework and analytical model.

The description of the research methods is too brief and does not provide specific research steps and data sources. It is recommended to provide a detailed description of the research methods, including the data collection and processing process, model establishment and validation process, etc., to help readers better understand and evaluate the reliability of the research.

The description of the results analysis is not clear, and the interpretation and explanation of the charts and data are insufficient. It is suggested to add detailed explanations and analysis of the charts and data in the results analysis section to better support the conclusions and arguments.

The conclusion is too brief and does not effectively summarize the main points and highlights of the paper. It is recommended to provide a more detailed summary of the research findings, indicating their significance and contributions, while also discussing future research directions and value.

It is necessary to further improve and refine the logical structure, theoretical framework, research methods, and results analysis section to enhance the quality and reliability of the paper.

The paper includes seven funding projects. Are all these funding projects relevant to the research content?

Reviewer #2

The study introduces an intriguing application of physics concepts in economics and management, providing valuable insights on enhancing sustainability in the high-tech industry and its development. The article's immersive reading experience is enhanced by the colorful and vivid language utilized.

However, it is important to point out certain flaws:

1. Lines 89-90 would benefit from including the initial source: Cao HX. Shell (Jieke) phenomenon and its scientific frame. Science and Technology Report No. 88005. Chinese Academy of Meteorological Science. SMA. 1988.

2. The functions of the boundary are indistinct and require clarification. The listed information such as “particularly the role of boundary shells in regulating and managing the exchange of matter, energy, and information” (lines 98-103), “maintaining a sustainable and healthy ecological environment, highlighting the need to consider the interconnectedness of ecological, economic, and social factors in sustainable development.” (lines 111-114), lacks precision and clarity.

3. Figure 1, Figure 2: It is recommended that the authors explore the potential external or internal factors contributing to the observed fluctuations. The authors could speculate on it and provide further clarification in the Conclusion or Discussion section.

4. Despite the thorough presentation of the results and conclusion, it is crucial to compare the current findings with other theories, including the “traditional model”. I urge the authors to either expand the Conclusion or include a Discussion section.

5. To ensure clarity, eliminate duplications such as lines 91-92 and 100-101 and correct all references typos found in lines 38, 43, 46, 47, and so on. Additionally, the Reference list should follow the guidelines of the journal and includes DOIs when applicable.

Reviewers' comments:

Reviewer's Responses to Questions

**Comments to the Author**

1. Is the manuscript technically sound, and do the data support the conclusions?

Reviewer #1: Partly

Reviewer #2: Yes

2. Has the statistical analysis been performed appropriately and rigorously? 

Reviewer #1: No

Reviewer #2: Yes

3. Have the authors made all data underlying the findings in their manuscript fully available?

Reviewer #1: Yes

Reviewer #2: Yes

4. Is the manuscript presented in an intelligible fashion and written in standard English?

Reviewer #1: No

Reviewer #2: Yes

5. Review Comments to the Author

Reviewer #1: There is an incorrect citation in line 120, and the font inconsistency in line 163.

The introduction of the paper does not effectively introduce the topic and feels abrupt. It is recommended to include some background information in the introduction, such as the importance of the high-tech industry and the significance of sustainable development, to provide a better foundation for the subsequent discussion.

The theoretical framework section lacks detailed exposition and explanation. It is suggested to delve deeper into the relationship between the Boundary Shell system's openness and sustainable development, and provide a clearer theoretical framework and analytical model.

The description of the research methods is too brief and does not provide specific research steps and data sources. It is recommended to provide a detailed description of the research methods, including the data collection and processing process, model establishment and validation process, etc., to help readers better understand and evaluate the reliability of the research.

The description of the results analysis is not clear, and the interpretation and explanation of the charts and data are insufficient. It is suggested to add detailed explanations and analysis of the charts and data in the results analysis section to better support the conclusions and arguments.

The conclusion is too brief and does not effectively summarize the main points and highlights of the paper. It is recommended to provide a more detailed summary of the research findings, indicating their significance and contributions, while also discussing future research directions and value.

It is necessary to further improve and refine the logical structure, theoretical framework, research methods, and results analysis section to enhance the quality and reliability of the paper.

The paper includes seven funding projects. Are all these funding projects relevant to the research content?

Reviewer #2: The study introduces an intriguing application of physics concepts in economics and management, providing valuable insights on enhancing sustainability in the high-tech industry and its development. The article's immersive reading experience is enhanced by the colorful and vivid language utilized.

However, it is important to point out certain flaws:

1. Lines 89-90 would benefit from including the initial source: Cao HX. Shell (Jieke) phenomenon and its scientific frame. Science and Technology Report No. 88005. Chinese Academy of Meteorological Science. SMA. 1988.

2. The functions of the boundary are indistinct and require clarification. The listed information such as “particularly the role of boundary shells in regulating and managing the exchange of matter, energy, and information” (lines 98-103), “maintaining a sustainable and healthy ecological environment, highlighting the need to consider the interconnectedness of ecological, economic, and social factors in sustainable development.” (lines 111-114), lacks precision and clarity.

3. Figure 1, Figure 2: It is recommended that the authors explore the potential external or internal factors contributing to the observed fluctuations. The authors could speculate on it and provide further clarification in the Conclusion or Discussion section.

4. Despite the thorough presentation of the results and conclusion, it is crucial to compare the current findings with other theories, including the “traditional model”. I urge the authors to either expand the Conclusion or include a Discussion section.

5. To ensure clarity, eliminate duplications such as lines 91-92 and 100-101 and correct all references typos found in lines 38, 43, 46, 47, and so on. Additionally, the Reference list should follow the guidelines of the journal and includes DOIs when applicable.

6. PLOS authors have the option to publish the peer review history of their article (what does this mean?). If published, this will include your full peer review and any attached files.

Reviewer #1: No

Reviewer #2: No

---

## [Author Response · Author response to Decision Letter 0]

6 Jan 2024

We would like to express our sincere gratitude for your constructive comments and suggestions. Your input has greatly enhanced the quality and presentation of our manuscript. We have carefully addressed all your concerns and provided detailed responses in the RESPONSES LETTER.

---

## [Decision Letter · Decision Letter 1]

22 Jan 2024

Does the openness of the Boundary Shell system influence the sustainable development of the high-tech industry?

PONE-D-23-29413R1

Dear Dr. YiMing Shi,

We’re pleased to inform you that your manuscript has been judged scientifically suitable for publication and will be formally accepted for publication once it meets all outstanding technical requirements.

Kind regards,

Voxi Heinrich Amavilah, Ph.D.

Academic Editor

PLOS ONE

Additional Editor Comments (optional):

Please address the following:

1. Some tables still contain some errors, please double-check carefully.

2. The conclusion section still lacks a comparison of the obtained results with those obtained from other theories.

3. The reference list and in-text citations (lines 173, 174, 177, 181, etc.) still do not follow the journal's guidelines.

Reviewers' comments:

Reviewer's Responses to Questions

**Comments to the Author**

1. If the authors have adequately addressed your comments raised in a previous round of review and you feel that this manuscript is now acceptable for publication, you may indicate that here to bypass the “Comments to the Author” section, enter your conflict of interest statement in the “Confidential to Editor” section, and submit your "Accept" recommendation.

Reviewer #1: All comments have been addressed

Reviewer #2: All comments have been addressed

2. Is the manuscript technically sound, and do the data support the conclusions?

Reviewer #1: Yes

Reviewer #2: Yes

3. Has the statistical analysis been performed appropriately and rigorously? 

Reviewer #1: Yes

Reviewer #2: Yes

4. Have the authors made all data underlying the findings in their manuscript fully available?

Reviewer #1: Yes

Reviewer #2: Yes

5. Is the manuscript presented in an intelligible fashion and written in standard English?

Reviewer #1: Yes

Reviewer #2: Yes

6. Review Comments to the Author

Reviewer #1: I appreciate the hard work of the authors in revising this manuscript. This latest revised version of the manuscript represents a significant quality improvement compared to the earliest version.

Some tables still contain some errors, please double-check carefully.

Reviewer #2: 1. The conclusion section still lacks a comparison of the obtained results with those obtained from other theories.

2. The reference list and in-text citations (lines 173, 174, 177, 181, etc.) still do not follow the journal's guidelines.

7. PLOS authors have the option to publish the peer review history of their article (what does this mean?). If published, this will include your full peer review and any attached files.

Reviewer #1: No

Reviewer #2: No

---

## [Editor Report · Acceptance letter]

8 Feb 2024

PONE-D-23-29413R1 

PLOS ONE

Dear Dr. Shi, 

I'm pleased to inform you that your manuscript has been deemed suitable for publication in PLOS ONE. Congratulations! Your manuscript is now being handed over to our production team.

Kind regards, 

on behalf of

Dr. Voxi Heinrich Amavilah 

Academic Editor

PLOS ONE